# Histone H2A variants alpha1-extension helix directs RNF168-mediated ubiquitination

Jessica L. Kelliher[1], Kirk L. West[1], Qingguo Gong[2] & Justin W. C. Leung [1✉]

Histone ubiquitination plays an important role in the DNA damage response (DDR) pathway. RNF168 catalyzes H2A and H2AX ubiquitination on lysine 13/15 (K13/K15) upon DNA damage and promotes the accrual of downstream repair factors at damaged chromatin. Here, we report that RNF168 ubiquitinates the non-canonical H2A variants H2AZ and macroH2A1/2 at the divergent N-terminal tail lysine residue. In addition to their evolutionarily conserved nucleosome acidic patch, we identify the positively charged alpha1-extension helix as essential for RNF168-mediated ubiquitination of H2A variants. Moreover, mutation of the RNF168 UMI (UIM- and MIU-related UBD) hydrophilic acidic residues abolishes RNF168-mediated ubiquitination as well as 53BP1 and BRCA1 ionizing radiation-induced foci formation. Our results reveal a juxtaposed bipartite electrostatic interaction utilized by the nucleosome to direct RNF168 orientation towards the target lysine residues in proximity to the H2A alpha1-extension helix, which plays an important role in the DDR pathway.

---

[1] Department of Radiation Oncology, College of Medicine, University of Arkansas for Medical Sciences, Little Rock, AR 72205, United States. [2] Hefei National Laboratory for Physical Science at the Microscale, School of Life Sciences, University of Science and Technology of China, 96 Jinzhai Road, 230027 Hefei, Anhui, China. ✉email: jwleung@uams.edu

Histone ubiquitination plays a pivotal role in the response to DNA damage[1]. Histone proteins H2A and H2AX are ubiquitinated by RNF168 at the N-terminal tail lysine (K) 13/15, RING1B-BMI1 at the C-terminal tail K118/119 and BRCA1/BARD1 at K127/129[2–6]. In particular, RNF168 is important in the double-strand break (DSB) repair process, which is demonstrated by its association with RIDDLE (Radiosensitivity, ImmunoDeficiency, Dysmorphic features, and LEarning difficulties) syndrome[7]. In DSB repair, RNF168-mediated H2A and H2AX K15 ubiquitination recruits 53BP1 to damaged chromatin via direct protein-protein interaction[5,8–10]. 53BP1 accrual at DNA breaks dictates DSB repair pathway choice by suppressing DNA end-resection[11]. Notably, RNF168 inactivation abolishes the recruitment of DNA repair proteins including BRCA1, RAD18, and RAP80 to DNA DSBs[8,12], which highlights the important role of RNF168 as a mediator in the assembly of DNA repair proteins at damaged chromatin.

Recent work provided important mechanistic insights into the regulation of RNF168 activity and how RNF168 orchestrates the DNA damage response (DDR) signaling pathway. RNF168 is a 571-residue protein composed of an E3 catalytic domain and multiple structural motifs that are required for its functions. The RNF168 catalytic RING domain is unique for its activity as replacing it with the RNF8 RING domain fails to restore its function in RIDDLE cells[13]. In addition, RNF168 has multiple ubiquitin-binding domains (UBD) including two MIU (Motif Interacting with Ub), two LR motifs, and a UMI (UIM [ubiquitin interacting motif]—and MIU [motif interacting with ubiquitin]-related UBD) motif. The C-terminal MIU2-LRM2 motifs (i.e., ubiquitin-dependent DSB recruitment modules 2 or UDM2) are primarily required for RNF168 ionizing radiation-induced foci (IRIF) formation via binding of the RNF168-dependent ubiquitin signal. On the other hand, the LRM1-UMI-MIU1 composes the UDM1 that is responsible for the RNF8-initiated RNF168 recruitment[7,9,14,15]. These motifs are functionally characterized in the context of IRIF formation and their ability to bind ubiquitin[14,16].

H2A is the most diverse histone family, comprised of 26 genes with four sub-families of non-canonical variants including H2AX, H2AZ, macroH2A, and H2A.Bbd. H2A variants nucleosomes occupy distinct locations in genomes[17]. In general, H2AZ is found in both euchromatin and heterochromatin. It occupies individual nucleosomes surrounding the transcription start site[18–20] and its level is relatively lower in transcribed regions[17]. Emerging evidence demonstrates that H2AZ is involved in the regulation of the DDR pathway, particularly in DSB repair[21–24]. A recent report showed that RNF168 ubiquitinates H2AZ[25], however, the molecular genetics of how H2AZ is ubiquitinated by RNF168 are unknown. MacroH2A was originally identified at the inactive X chromosome and is linked to heterochromatin[26,27], but not required for X inactivation[28]. Recent studies revealed an important role of macroH2A in DSB repair[29,30], base excision repair[31], replicative protection[32,33] and telomere lengthening[34]. Overall, non-canonical H2A variants are implicated in maintaining genome stability.

RNF168 is a crucial player in the DDR pathway. Therefore, understanding the mechanistic action of RNF168 is important for identifying specific substrates and developing a therapeutic strategy to treat DNA repair-related genetic diseases by harnessing the DDR pathway. The mechanism of RING finger E3 ligase ubiquitination catalysis is well established[35–37]. Interestingly, the target specificity is not always conferred by the RING-domain of the E3 ligase. Several RING finger E3 ligases require additional recognition elements, either from a binding partner or a motif outside of the RING domain, to target their substrates[38–40].

There are several studies showed biochemical and structural evidence of how RNF168 and RING1B engage the nucleosome via interaction with the H2A/H2B acidic patch. This interaction is established through the electrostatic interaction between the negatively charged acidic patch on the nucleosome and the positively charged arginine-rich helix on the E3 ligases[41,42]. NMR and computational modeling revealed the details of the molecular specificity between the RNF168-RING, RNF8-RING domains and the nucleosome. It also defined the RNF168 arginine anchor as important for interaction with the nucleosome acidic patch and identified H2B E110 as a requirement for RNF168 target specificity[43]. Nevertheless, this interaction does not fully explain how RNF168 and RING1B choose the specific target lysine residue among the nine lysine residues on the H2A tails or the many lysine residues on the nucleosomal surface. Additional regulation may exist to direct the precise spatial control within this RNF168-mediated ubiquitination.

In this study, we sought to dissect the regulatory mechanism of RNF168 catalytic activation through understanding the molecular basis of RNF168-substrate recognition specificity. We focus on both nucleosome and RNF168 structural motifs. Our results indicate that atypical H2A variants are substrates of RNF168. We show that RNF168 catalyzes site-specific lysine ubiquitination on each H2A variant within the unstructured divergent N-terminus. An additional regulatory mechanism at the H2A variants alpha1-extension helices of H2A variants promotes RNF168 orientation towards the target lysine residue on the nucleosome. Our findings provide potential missing links between RNF168 and the downstream effectors and highlight the complexity of the nucleosome structure in the RNF168-mediated DDR pathway in the context of chromatin.

## Results

**Atypical H2A variants are bona fide RNF168 substrates**. The H2A family contains diverse atypical variants, most of them have lysine-rich histones tails (Fig. 1 and supplementary Fig. 1). RNF168 catalyzes H2A and H2AX site-specific ubiquitination at K13/K15 residues[5]. We ask the question of whether the atypical H2A variants are also RNF168 substrates (Fig. 1a). Since RNF168 is the limiting factor in the DDR pathway, the basal level of target ubiquitination is low[5,41]. To examine whether RNF168 can target atypical H2A variants, we ectopically expressed myc-RNF168 and SFB (S-protein, Flag, Streptavidin binding peptide)-H2A variants in HEK293T cells. With RNF168 overexpression, we observed increased ubiquitination in H2AZ, macroH2A1, and macroH2A2, but not H2A.Bbd (Fig. 1b). Except for H2A.Bbd, which does not contain lysine, these H2A variants have multiple divergent lysine residues on both N-termini and C-termini that are not conserved with H2A and H2AX (Fig. 1a and supplementary Fig. 1). To determine whether RNF168 ubiquitinates these variants directly, we performed in vitro ubiquitination assay using H2AZ and macroH2A-containing nucleosomes. Consistent with a previous report[25], H2AZ is a bona fide RNF168 substrate (Supplementary Fig. 2a). In addition, our data show that RNF168 also ubiquitinates macroH2A1 in vitro (Fig. 1c-e).

In order to profile the H2A variants ubiquitination at their histone tails and map the RNF168-targeted residues, we separated the histone tail lysine residues by mutating either the N-terminal or C-terminal lysine (K) residues to arginine (R) (N-5K to R and C-4K to R) on H2AZ and (N-4K to R and C-4K to R) on macroH2A1/2. The K to R conversion abrogated the ubiquitination target while maintaining the charge properties of the physical structure. C-4KR mutants dramatically reduced the ubiquitination levels in H2AZ and macroH2A1/2 (Supplementary Fig. 2b-d). Interestingly, RNF168 overexpression catalyzes H2AZ and

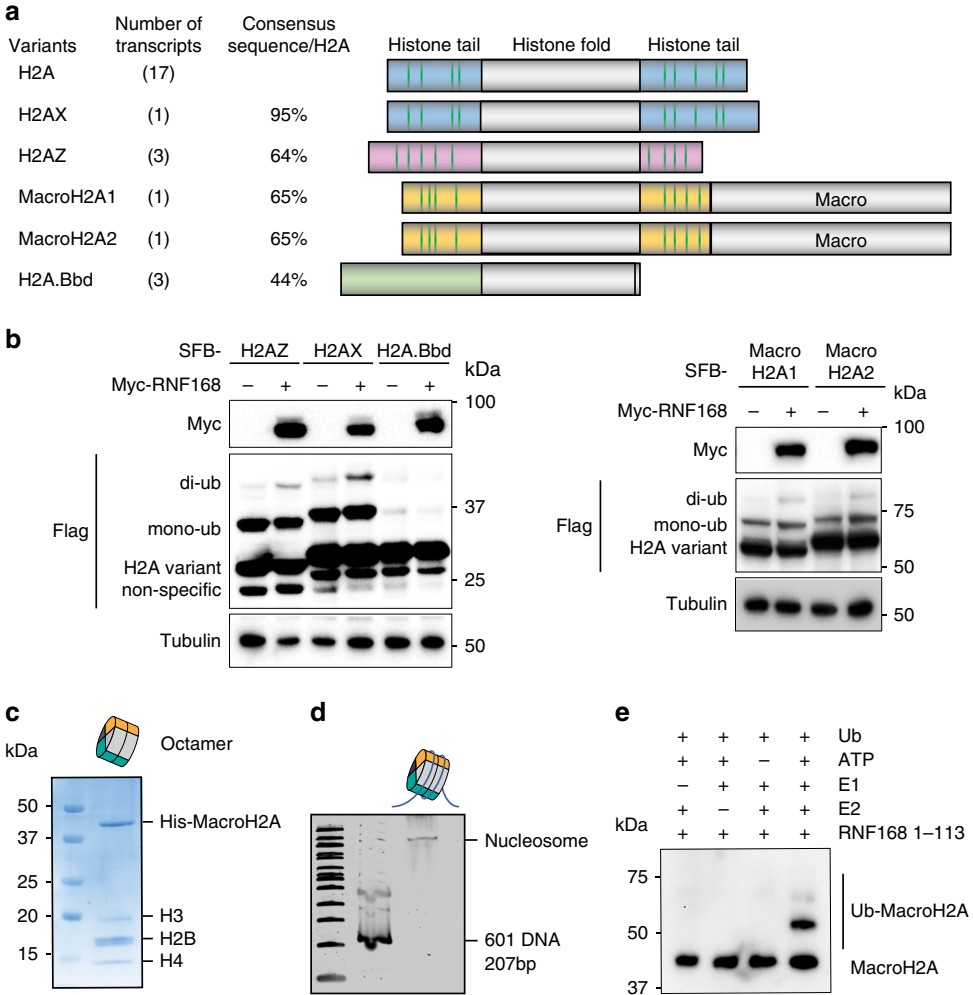

**Fig. 1 H2AZ and macroH2A1/2 are RNF168 substrates. a** Schematic illustration of the human H2A family. Consensus sequences were analyzed in comparison to H2A. For macroH2A1 and macroH2A2, the macro domain is excluded from the sequence homology analyses. Green lines represent lysine residues in the histone tail regions. **b** RNF168 ubiquitinates H2A variants. Cells were co-transfected with Myc-RNF168 and SFB-H2A variants in HEK293T cells, then harvested in SDS-PAGE sample buffer followed by western blot analysis with indicated antibodies. H2AX was used as a positive control. Repeated three times independently with similar results. **c** Coomassie staining of macroH2A-containing octamer. **d** Analysis of in vitro reconstitution of macroH2A-containing nucleosome core particles (NCPs). The 207 bp 601 DNA fragment was analyzed alone or after NCP reconstitution. 100 bp DNA ladder indicates size. **e** in vitro ubiquitination assay of macroH2A-conating nucleosome was incubated with E1, E2, RNF168 (1-113), ATP and ubiquitin in 1× ubiquitination buffer at 30 °C overnight. The reactions were stopped by adding 2× SDS sample buffer. Samples were analyzed by western blot and macroH2A antibody. Repeated three times independently with similar results. Source data are provided as Source Data file.

macroH2A1/2 ubiquitinations in the C-terminal K to R mutant but not the N-terminal K to R mutant (Supplementary Fig. 2b-d). These data suggest that in the atypical H2A variants the C-terminus is the major ubiquitin acceptor and the N-terminus is the RNF168 target similar to H2A and H2AX.

To pinpoint the major ubiquitination sites and the RNF168-targeted lysine, we systematically generated lysine to arginine (KR) mutations on each H2AZ lysine residue (Fig. 2a). Interestingly, none of the mutations show a significant reduction in ubiquitination level, suggesting that there may be more than one major ubiquitin acceptor (Supplementary Fig. 2e). In parallel, we generated a single lysine-only platform by reversing each arginine to lysine in the H2AZ-9KR mutant (e.g., 9K to R-R4K). In agreement with our previous observation (Supplementary Fig. 2b) that the H2AZ C-terminus is the major ubiquitin acceptor, H2AZ 9K to R-R120K, 9K to R-121K, and 9K to R- R125K display a strong basal level of mono-ubiquitination compared to other single lysine H2AZ 9K to R-K mutants (Fig. 2b). Notably, co-expressing RNF168 failed to ubiquitinate H2AZ lacking K15 (Fig. 2c) but it

was able to ubiquitinate the H2AZ-9K to R-R15K, suggesting that RNF168 catalyzes H2AZ ubiquitination at K15 specifically (Fig. 2d). Using a similar approach, we identified K11 as the RNF168 target lysine residue on macroH2A1 and 2 (Fig. 2e, f). Given that the N-terminal ubiquitination of H2A variants is substantially lower than that of the C-terminal ubiquitination, and the basal level of RNF168-mediated ubiquitination is low, it is difficult to measure the change of H2A-ubiquitinations in RNF168-depleted cells. In order to examine the effect of RNF168 in H2AZ and macroH2A ubiquitination, we depleted RNF168 using siRNAs in HEK293T cells stably-expressing H2AZ 9K to R-R15K or macroH2A1/2 8K to R-R11K and performed a pull-down assay. Our data showed that RNF168 depletion reduces H2AZ and macroH2A1/2 ubiquitination at the specific lysine (K15 for H2AZ and K11 for macroH2A1/2) (Fig. 2g-i). Ectopic overexpression of the GFP-H2AZ 9K to R-R15K, GFP-macroH2A1 8K to R-R11K and GFP-macroH2A2 8K to R-R11K did not affect the formation of ubiquitin and 53BP1 accrual at DNA damage sites (Supplementary Fig. 3a, b).

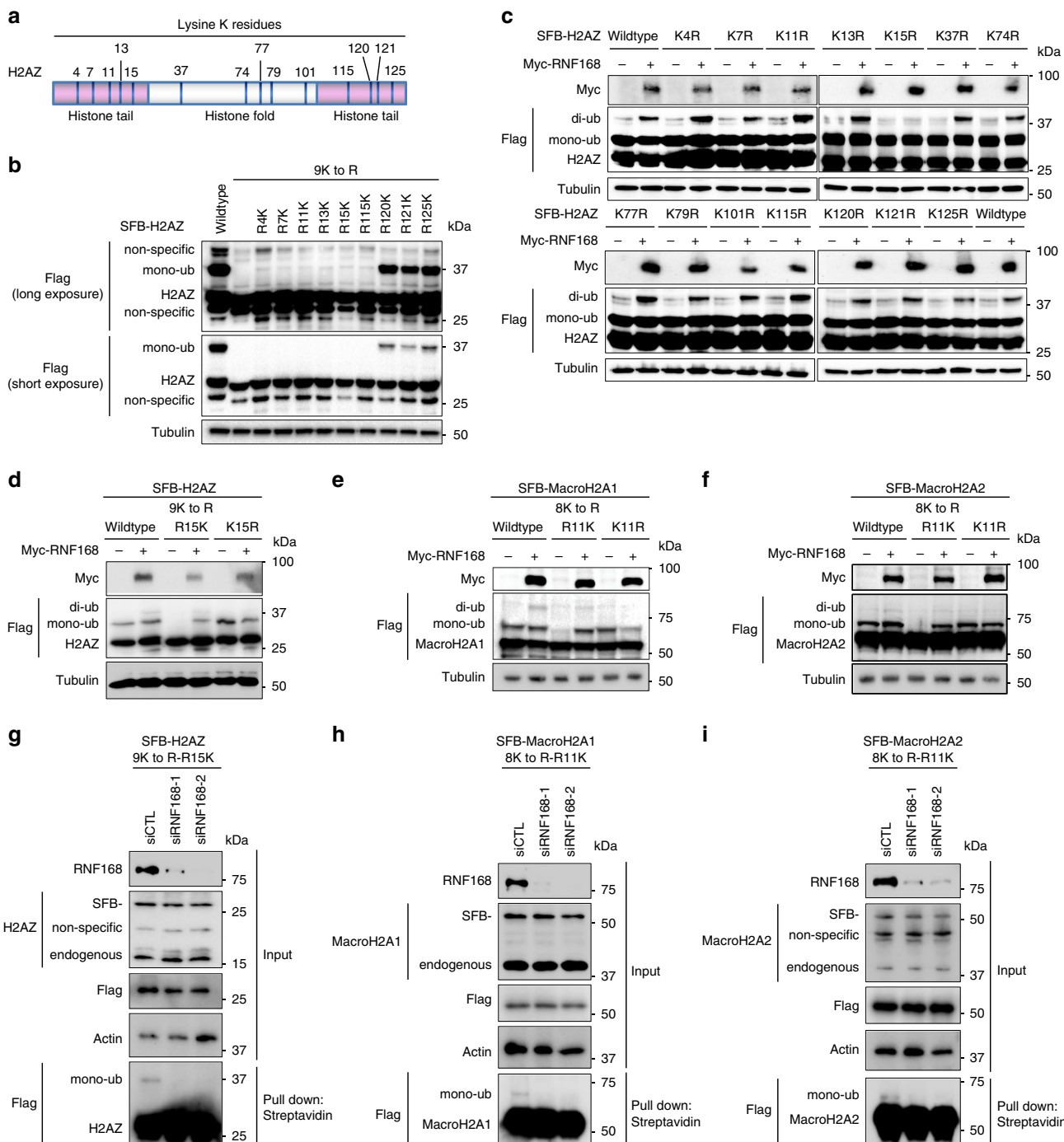

**Fig. 2 RNF168 ubiquitinates H2A variants at a specific lysine residue. a** Schematic diagram of the lysine residue distribution of H2AZ. **b** H2AZ C-terminus is the major Ub-acceptor. SFB-H2AZ wildtype and mutants were transfected in HEK293T cells for 24 h, harvested with SDS-PAGE sample buffer, and followed by western blot analysis using Flag antibody and tubulin as loading control. Repeated two times independently with similar results. **c** RNF168 specifically ubiquitinates H2AZ at K15. Co-transfection of Myc-RNF168 and SFB-H2AZ wildtype and mutants with lysine to arginine mutation in HEK293T cells as indicated. Repeated three times independently with similar results. **d–f** RNF168 ubiquitinates H2A variants at a specific lysine residue. SFB-H2AZ, SFB-macroH2A1, and SFB-macroH2A2 wildtype and mutants were co-transfected with Myc-RNF168 in HEK293T cells followed by western blot analysis using antibodies as indicated. Repeated at least three times independently with similar results. **g–i** RNF168 is required for site-specific ubiquitination of H2A variants. HEK293T cells with stable expression of CMV-SFB-H2AZ 9K to R-R15K, CMV-SFB-MacroH2A1 8K to R-R11K, EF1α -SFB-MacroH2A2 8K to R-R11K were transfected with siRNAs targeting RNF168. Cells were harvested and pulled down using streptavidin agarose. Pull-down samples were analyzed by western blot with indicated antibodies. Repeated four times for g-h and two times for i with similar results. Source data are provided as Source Data file.

RNF168-mediated H2A and H2AX are DNA damage-induced[41]. Here, we test whether RNF168-dependent H2AZ and macroH2A1/2 ubiquitinations are DNA damage induced. We stably expressed H2AZ and macroH2A1/2 mutants with only the RNF168-targeted lysine (SFB-H2AZ 9K to R-R15K, SFB-macroH2A 8K to R-R11K and SFB-macroH2A 8K to R-R11K) in cells. After irradiation, we detected an increase in ubiquitination of the RNF168-specific lysine (Supplementary Fig. 4a-c). Collectively, these data demonstrate that the site-specific ubiquitination of H2A variants is directly mediated by RNF168 upon DNA damage.

### RNF168-mediated ubiquitination requires H2A acidic patches.
To determine how RNF168 selects these lysine targets on H2A variants, we first looked for the conserved structural element among the H2A variants that is required for RNF168-mediated ubiquitination. Sequence alignment showed that H2AZ and MacroH2A1/2 contain intact acidic patches similar to those in H2A and H2AX (Fig. 3a) which are required for RNF168-mediated K13/15 ubiquitination[41,42]. These acidic patches structurally reside on the same surface of the nucleosome (Fig. 3b), we hypothesize that the H2AZ and macroH2A1/2 acidic patches are also involved in regulating their ubiquitination. By mutating the key aspartic acid corresponding to H2A D90 within the acidic patches, H2AZ (D93A) and macroH2A1/2 (D87A) ubiquitinations are drastically reduced even though RNF168 is not the limiting factor in cells (Fig. 3c-e). These results are similar to H2A and H2AX acidic patch mutation[41,42] and suggest that this well-conserved structural entity on atypical H2A variants is also required for docking RNF168 and other E3 ligases on the nucleosome.

### RNF168 ubiquitination requires H2A alpha1-extension helices.
Although these H2A variants have an acidic patch that is indispensable for their ubiquitination, we speculated that there is an additional layer of regulation for fine-tuning the RNF168 target selectivity on the lysine-rich nucleosome histone tails besides the distal acidic patch. Intriguingly, the N-terminal tail sequences of H2A variants, including the RNF168-targeted lysine residues, are poorly conserved (Fig. 4a). Analysis of 13 different crystal structures of human H2A-containing nucleosomes (Supplementary Fig. 5a-c) and H2A-containing nucleosomes from 5 different species (Supplementary Fig. 5d-f) showed structural and

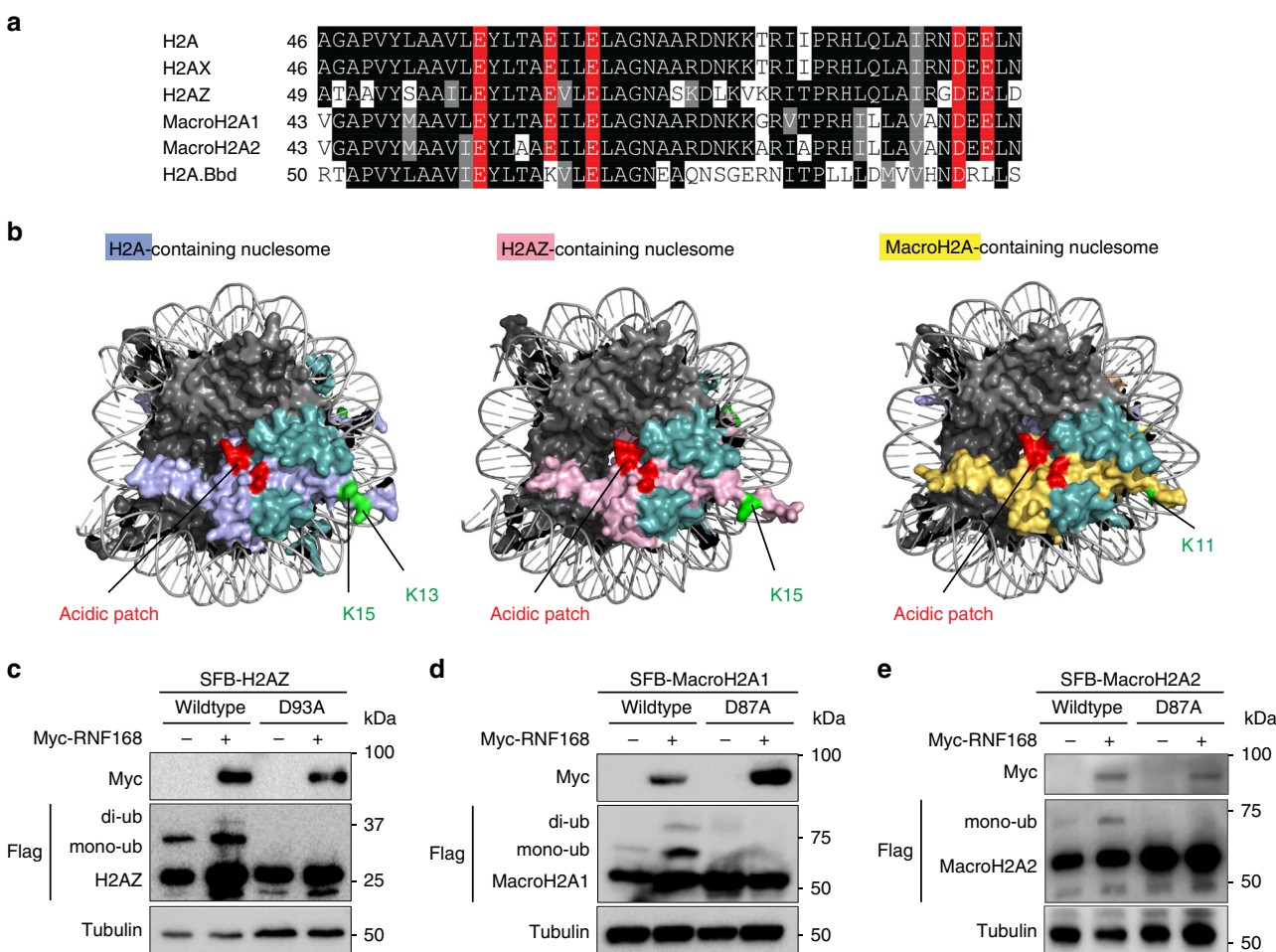

**Fig. 3 Mutation of the acidic patches of H2A variants impairs ubiquitination. a** H2A variants have an evolutionary conserved acidic patch. The graphic represents sequence alignment of H2A variants. Acidic residues required to form an intact acidic patch were highlighted in red. **b** Structural illustration of the H2A-containing, H2AZ-containing and MacroH2A-containing nucleosome. H4 is in light gray, H3 is in dark gray, H2B is in teal, and H2A variants are in the following colors: H2A–light blue; H2AZ–pink; MacroH2A–yellow. The evolutionary and structurally conserved acidic patch is in red and the RNF168-targeted lysine residues are in green. **c–e** The acidic patch of H2A variants is required for RNF168-mediated site-specific ubiquitination. HEK293T cells were transfected with Myc-RNF168 and SFB-H2A variants and their acidic patch mutants (H2AZ-D93A; MacroH2A1-D87A; MacroH2A2-D87A) as indicated. Cells were harvested 24 h after transfection and ubiquitinations were analyzed by western blot with indicated antibodies. Repeated three times independently with similar results. Source data are provided as Source Data file.

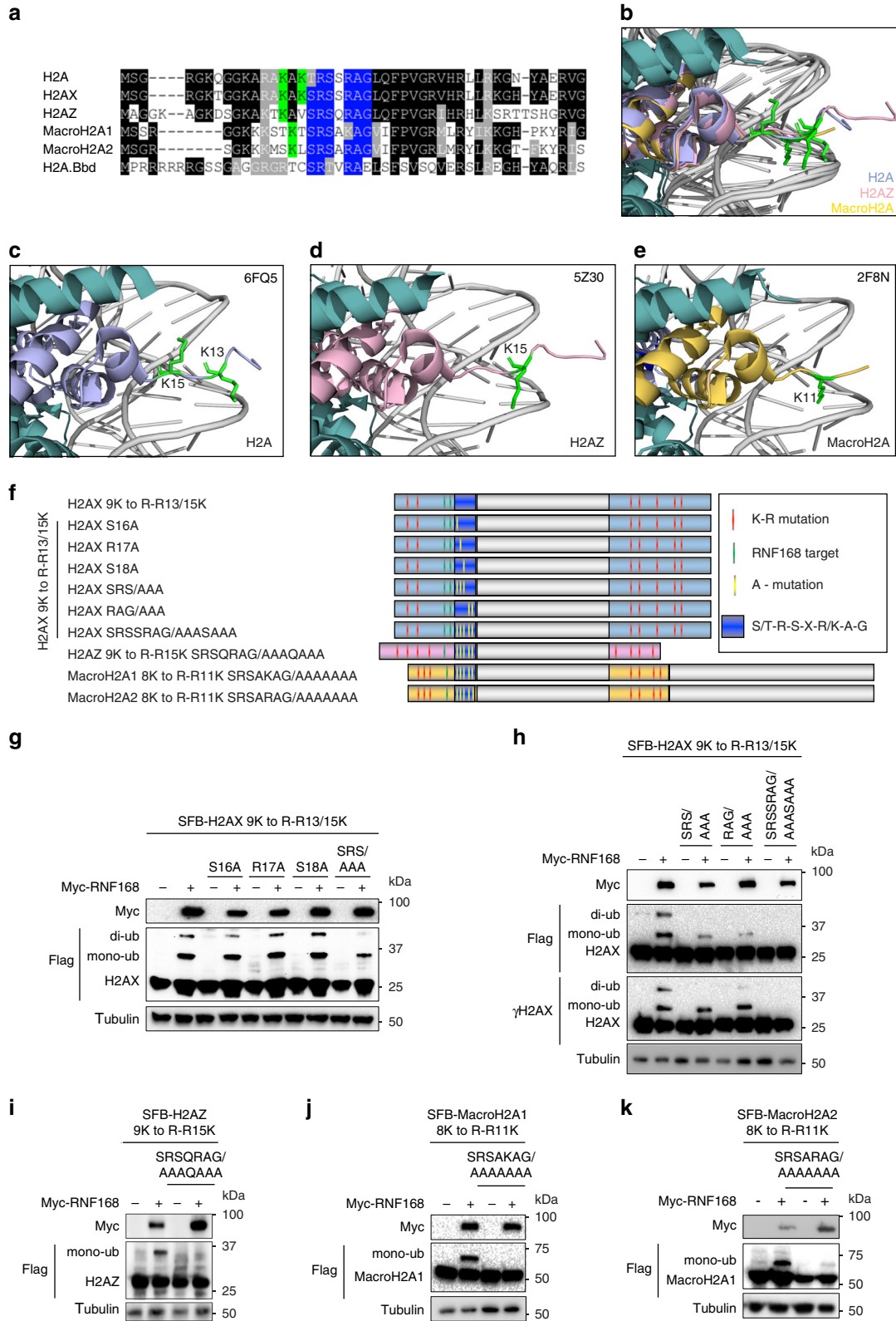

positional consistency at the N-terminal K13 and K15 of H2A. Interestingly, superimposition of H2A, H2AZ, and macroH2A containing nucleosomes illustrated positional differences on the RNF168-targeted lysine side chain (Fig. 4b-e). This observation provided an opportunity to map the elements that endow the RNF168 target specificity from multiple lysines residing at the N-terminus of H2A variants.

Using sequence alignment analysis, we observed that the alpha1-extension region on the N-terminal tails is evolutionarily conserved among these H2A variants (Fig. 4a, highlighted in blue). More importantly, this region is the only consensus sequence in the proximity of the RNF168-targeted lysine residues outside of the histone fold domains. The alpha1-extension helix contains seven amino acid residues S/T-R-S-X-R/K-A-G. Within

**Fig. 4 RNF168-mediated H2A ubiquitination requires the histone tail alpha1-extension helix. a** Sequence alignment of the N-terminal tail and the alpha-1 extension helices of H2A variants. Residues highlighted in blue represent the evolutionary conserved alpha1-extension helix and those in green represent the RNF168-targeted lysine residues. **b** Analysis of the superimposition of H2A-containing, H2AZ-containing, and macroH2A-containing nucleosome, RNF168-targeted lysine residues were labeled green. **c–e** Individual H2A-containing (PDB:6FQ5), H2AZ-containing (PDB: 5Z30), and macroH2A-containing (2F8N) nucleosomes at their N-terminal tails. H2B is colored in teal, H2A is in light blue, H2AZ is in pink, and MacroH2A is in yellow. The RNF168-targeted lysines sidechains were labeled green and presented as a stick. **f** Schematic illustration of the mutants used are presented as in **g–k**. **g–h** Conserved residues on alpha1-extension helix are required for RNF168-mediated H2AX ubiquitination. Myc-RNF168 and SFB-H2AX alpha-1 extension helix mutants as indicated were co-transfected in HEK293T cells followed by western blot analysis. Repeated three times independently with similar results. **i–k** Alpha1-extension helix is required for RNF168-dependent ubiquitination of H2A variants. SFB-H2A variants and their alpha1-extension helix mutants were co-transfected with Myc-RNF168 in HEK293T cells. Cells were harvested 24 h after transfection and analyzed by western blot. Repeated three times with similar results. Source data are provided as Source Data file.

this region, the fourth residue is not conserved among H2A variants and the sixth alanine residue is buried inside the nucleosome and should not affect the binding surface for RNF168 engagement. Therefore, we predict that these two residues are not essential for RNF168 activity towards the target lysine residues. By alanine scanning mutagenesis of the alpha1-extension helix, we systematically generated mutants (Fig. 4f and Supplementary Fig. 6a) to examine their effects on RNF168-mediated ubiquitination. Single-amino acid mutations on the first three amino acids, S16A, R17A, and S18A, of the alpha1-extension helix did not show any effect on the RNF168-mediated K13/K15 ubiquitination (Fig. 4g) while combinatory mutations on the SRS/AAA and RAG/AAA mutants drastically attenuated the RNF168-mediated H2AX K13/15 ubiquitination (Fig. 4h and Supplementary Fig. 6b). Strikingly, H2AX K13/K15 ubiquitination was not detectable in the SRSSRAG/AAASAAA mutants (Fig. 4h). Consistent with H2AX, RNF168-mediated H2AZ and macroH2A1/2 ubiquitinations were also abolished in the alpha1-extension helix mutants (Fig. 4i-k). Similar to H2AX K13/15R mutants, reconstitution of GFP-H2AX SRSSRAG/AAASAAA restored 53BP1 in H2AX KO cells[41] (Supplementary Fig. 6c). These data demonstrated that at least two structurally conserved regulatory elements, the acidic patch and the alpha1-extension helix, are required for RNF168 activity on the proximal lysine residues (Fig. 5a).

**Alpha1-extension helix dictates RNF168 target selectivity.** In addition to the role of the alpha1-extension helix in RNF168-mediated ubiquitination of H2A variants, we observed a proximity requirement for the RNF168-targeted lysine residues. The RNF168-targeted lysine residues in H2A variants are located three amino acids away from the first serine residue of the alpha1-extension helix. H2AZ has an additional lysine residing five amino acids from the alpha1-extension helix which is not ubiquitinated by RNF168 (Fig. 2c and Fig. 4a). To better understand the molecular requirements of target lysine positioning on RNF168 substrates, we used the *Saccharomyces cerevisiae* H2A(X) orthologue, yeast HTA (yHTA), which contains a lysine residue at the fourth amino acid upstream of the alpha1-extension helix motif, to test whether human RNF168 can catalyze the ubiquitination of yHTA (Fig. 5b). No discernible increase in ubiquitination observed in yHTA with RNF168 overexpression (Fig. 5c). Interestingly, after deletion of the three amino acids between the lysine and first serine of the alpha1-extension helix, RNF168 was able to ubiquitinate the mutant yHTA K13m. (Fig. 5c). We conclude that the proximity of the target lysine and the alpha1-extension helix is critical for RNF168 target recognition and selectivity.

**RNF168 functionally requires the UDM1 region.** Mutations of H2A N-terminal alpha1-extension helix abolishes RNF168-mediated target ubiquitination. Therefore, we speculate that this

sequence may contribute to RNF168 target recognition similar to the electrostatic interaction between the nucleosomal acidic patch and RNF168 arginine anchor[42,43]. To this end, we analyzed the electrostatic potential distribution on the H2A-containing, H2AZ-containing, and macroH2A-containing nucleosomes. Interestingly, we observed an opposite charge between the N-terminal region and the C-terminal region near the target lysine residues (Fig. 5d) in H2A-containing and H2A variant-containing nucleosomes. They consistently showed a negative electrostatic potential at their C-termini, while their N-termini were all electrostatically positive (Fig. 5d). This electrostatic potential discrepancy between the N-terminus and C-terminus may contribute to the engagement and orientation for protein, including RNF168, on the nucleosome surface. The positively charged H2A N-terminus may serve as an additional regulatory entity that is important for the specificity of the RNF168-mediated lysine ubiquitination, which leads to our hypothesis that there should be another regulatory sequence for RNF168 target specificity besides the arginine anchor.

Previous studies showed that the RNF168 a.a. 1-113 fragment (RING domain and arginine anchor) is sufficient to catalyze H2A and H2AX ubiquitination in vitro[41–43]. However, reconstitution of RNF168 a.a. 1-113 in RNF168 knockout (KO) cells did not restore 53BP1 IRIF formation (Fig. 6b), whereas reconstitution of RNF168 a.a. 1-190 (RING-UDM1 {LRM-UMI-MIU1} motifs) in RNF168 KO cells restored 53BP1 IRIF despite its inability to form IRIF (Fig. 6b). These data suggest that RNF168 LRM-UMI-MIU1 (UDM1) domain is required for H2A and H2AX ubiquitination in cells.

To further dissect the molecular regulation by which RNF168 catalytically triggers ubiquitination of H2A variants using its N-terminus RING-LRM-UMI-MIU1 domains, we performed a domain swapping experiment (Fig. 6a). Consistent with a previous report, the substitution of RNF168 RING-domain with RNF8 RING domain (RNF168 ΔRING-RNF8^RING) did not restore 53BP1 foci in RNF168 KO cells[13]. This suggests that the RING domain is required for RNF168 function (Fig. 6c), possibly through binding with a distinct E2 conjugating enzyme[13]. We then generated chimeric proteins by tethering RNF8 ΔRING with the RNF168 N-terminus residues 16-113 and 16-190 (Fig. 6a) to artificially localize these fragments to DNA damage sites. Surprisingly, even though RNF8 ΔRING-RNF168^16–113 can form foci, it does not restore 53BP1 IRIF in RNF168 KO cells (Fig. 6c). On the other hand, RNF8 ΔRING-RNF168^16–190 is able to restore 53BP1 IRIF in RNF168 KO cells similar to RNF168 1-190 (Fig. 6c). These results further confirm that the LRM-UMI-MIU1 motifs are essential for the RNF168-mediated DDR pathway, independent of RNF8.

**RNF168 UMI motif is required for H2A ubiquitination.** Previously characterized arginine loops extending from the RING domain are highly conserved across evolution (Supplementary

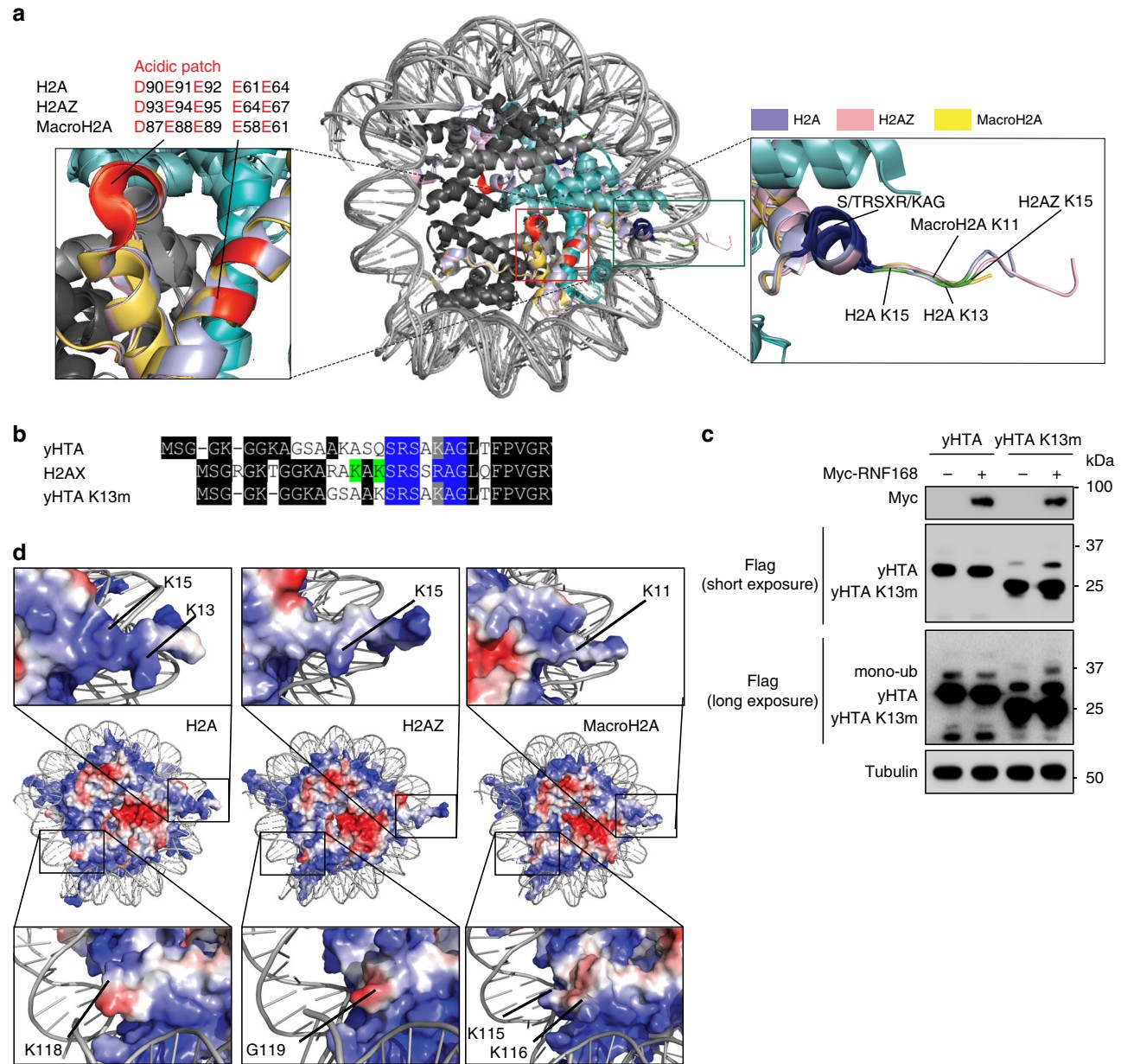

**Fig. 5 Ubiquitination requirement for RNF168 selectivity on H2A variants. a** Superimposition of H2A-containing, H2AZ-containing, and macroH2A-containing nucleosome illustrates the detailed residues required for RNF168-mediated site-specific ubiquitination. Zoom on the acidic patch (red), alpha1-extension helix region (blue) and RNF168-targeted lysine residues (green). **b** Proximity restriction of RNF168-mediated site-specific ubiquitination on H2A. Sequence alignment of human H2AX and yeast HTA (yHTA) at the N-terminal tail. The N-terminal sequence of the yHTA K13m mutant (ΔASQ) used as in **c**. The alpha1-extension helix residues are highlighted in blue. The proximal lysine residues are highlighted in green. **c** SFB-yHTA and mutant were co-transfected with Myc-RNF168 in HEK293T cells for 24 h and analyzed by western blot. Repeated three times independently with similar results. **d** Electrostatic potential (red-negative, blue-positive) analyses of H2A variant-containing nucleosomes. Zoomed illustration of the N-terminus and C-terminus of H2A variants. Histone tail lysines were labeled. H2AZ C-terminal lysines were absent in the illustration and G119 residue was marked. Source data are provided as Source Data file.

Fig. 7a). These basic arginine residues promote the docking of RNF168 on the acidic patch of the nucleosome[42,43]. Interestingly, the alpha1-extension helix includes two basic arginine residues and two serine residues, which potentially can contribute to hydrogen bonds formation between molecules. We speculate that the positively charged H2A N-terminal region (Fig. 5d), if not the H2A alpha1-extension helix specifically, may interact with negatively charged residues on RNF168. Intriguingly, we observed that RNF168 contains a negatively charged acidic residue-rich region between amino acids 96-177 within the LR-UMI-MUI motifs. With the limited structural information that is available

on the complete RING-LR-UMI-MUI fragment, it is challenging to predict the functional acidic region for RNF168 activity. To this end, we systematically generated six small internal deletion mutants based on the acidic residue clusters (Supplementary Fig. 7a). We then examined 53BP1 IRIF in U2OS-RNF168 KO cells reconstituted with GFP-RNF168 wildtype or mutants. Reconstitution of GFP-RNF168 in RNF168 KO cells restored 53BP1 and BRCA1 IRIF formation. Among the six deletion mutants, only GFP-RNF168 Δ143-144 reconstitution could not restore 53BP1 and BRCA1 IRIF in RNF168 KO cells (Fig. 7a-b and Supplementary Fig. 7b). The majority of GFP-RNF168

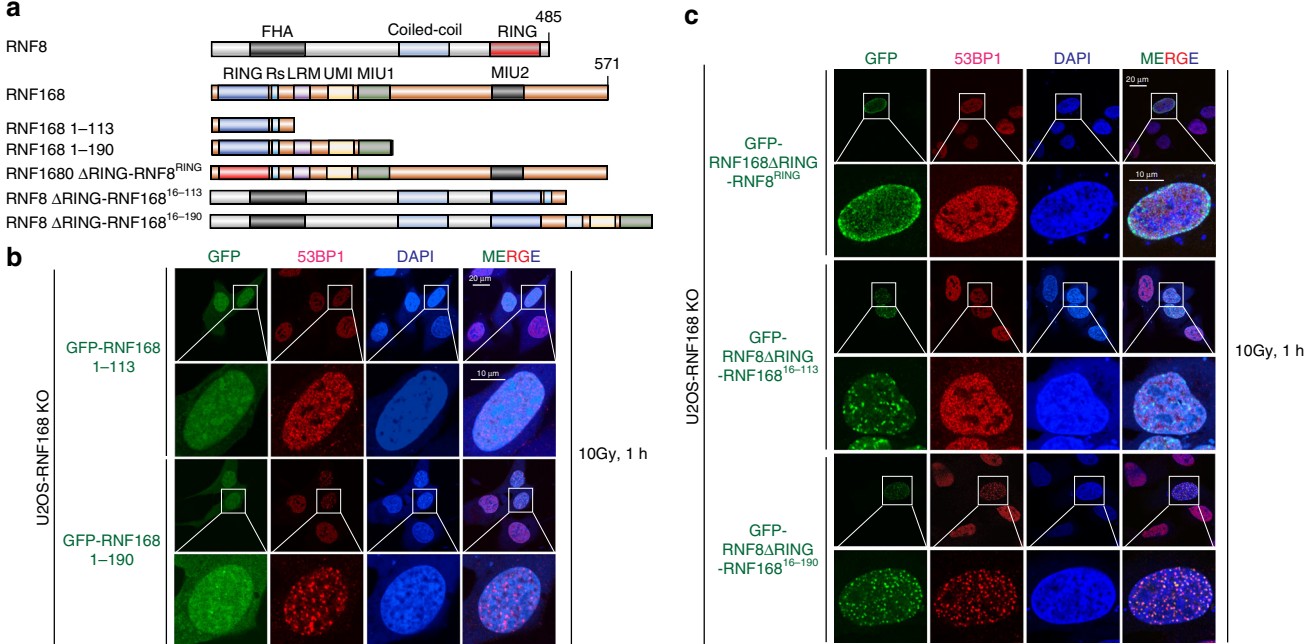

**Fig. 6 UDM1 is required for RNF168-mediated 53BP1 recruitment. a** Schematic illustration of RNF8 and RNF168 structural domain organization, RNF168 fragments, and chimeric proteins used as in **b** and **c**. FHA–forkhead-association domain; coiled-coil–coiled-coil domain; RING–ubiquitin E3 ligase RING domain; Rs-arginine anchor; LRM–LR motif; UMI–UIM-and MIU-related ubiquitin binding domain; MIU–motif interacting with ubiquitin. **b–c** Transient transfection of GFP-tagged RNF168 fragments and chimeric proteins in U2OS RNF168 KO cells. Cells were irradiated with 10 Gy and allowed to recover for 1 h followed by immunofluorescence analysis using 53BP1 antibody as indicated. Cells were counterstained with DAPI. Repeated at least two times independently with similar results. Source data are provided as Source Data file.

Δ143-144 cells displayed pan-nuclear localization or small punctuated foci after irradiation (Fig. 7b). Even though there was a small subset of GFP-RNF168 Δ143-144 that showed foci-like nuclear localization (Supplementary Fig. 7c), we did not see a restoration of 53BP1 IRIF or BRCA1 IRIF in the RNF168 KO cells with GFP-RNF168 Δ143-144 reconstitution (Fig. 7b and Supplementary Fig. 7c). Consistently, RNF168 Δ143-144 failed to ubiquitinate in H2AX K13/K15 (Fig. 7c). GFP-RNF168 Δ143-144 is recruited to laser-induced micro-irradiation similarly to GFP-RNF168 wildtype (Fig. 7d). These results suggested that the E143/E144 are important for RNF168-mediated ubiquitination without perturbing its DSB recruitment or its intrinsic activity[8,42]. Together, our data demonstrated a possible molecular mechanism in the regulation between the nucleosome and RNF168 in the DDR pathway.

To further determine whether the arginine anchor and acidic residues E143/E144 are required for RNF168 localization or orientation on nucleosome, we performed proximity ligation assay (PLA) using specific antibodies against GFP and γH2AX. Surprisingly, R57D and Δ143-144 mutants showed strong PLA positive signals with γH2AX similar to RNF168 wildtype. RNF168 R57D/ Δ143-144 double mutant displayed a dramatic reduction in PLA signal (Fig. 7e, f). Together, these data demonstrated that the RNF168 arginine anchor and E143/E144 acidic region are important for recruiting or positioning RNF168 on chromatin.

## Discussion

Histone ubiquitination is an integral part of the DDR pathway[1]. RNF168, as a key DDR E3 ligase, ubiquitinates H2A and H2AX at lysines 13 and 15 which recruits 53BP1 to damaged chromatin via its ubiquitination-dependent recruitment (UDR) motif[5,6,10]. Since RNF168 is the limiting factor within the DDR pathway, RNF168-mediated H2A and H2AX K13/K15 ubiquitination is relatively weak at the basal level[5,41]. In this study, using ubiquitination and

53BP1 IRIF formation as readouts, we utilized a simple cell-based genetic approach by co-expressing RNF168 and H2A variants to determine the regulatory elements on both E3 ligase and substrates. Notably, RNF168 is also required for the recruitment of other DDR proteins, including BRCA1, RAP80, and RAD18, to DNA damage sites[7]. One possibility is that there are yet-identified RNF168 substrates regulating the DDR pathway. Our study uncovers H2A variants H2AZ, macroH2A1, and macroH2A2 as targets for DNA-damage induced RNF168 ubiquitination, which may potentially provide additional docking platforms for recruiting downstream DNA repair proteins to DNA breaks. Our findings also reveal an additional mechanism of how RNF168 selectively recognizes the target lysine.

Among the core histones, the H2A family has the highest sequence divergence, which is encoded by 26 genes, 17 of which are classified as canonical H2A and 9 as atypical H2A[44]. Despite the poor conservation of the H2A variant histone tails, they are all predominately ubiquitinated at the C-terminus possibly by RING1B/BMI1[45,46]. Interestingly, SKP2, BRCA1, and CULLIN3 were also identified as ubiquitin ligases for macroH2A at the C-terminus[32,46–48]. In this study, we discover RNF168 ubiquitinates specific lysine residues at the N-termini of H2AZ (K15) and macroH2A1/2 (K11) (Fig. 2d-f). While the N-terminal tails of H2A variants share low sequence homology, the alpha1-extension helix is highly evolutionarily conserved. Comparison of the tails of histone variant, despite their flexibility and dynamic nature, provides insight into the ubiquitinat-able proximity of the target lysine from the conserved region.

Emerging genetic and structural evidence demonstrates that RNF168 engages with the nucleosome surface by employing the arginine anchor onto the H2A/H2B acidic patch[43,49]. This electrostatic interaction is quintessential for multiple chromatin factors and enzymes to target the unique acidic patch on the nucleosome surface[50]. Notably, RNF168 is a DDR E3 ligase that is

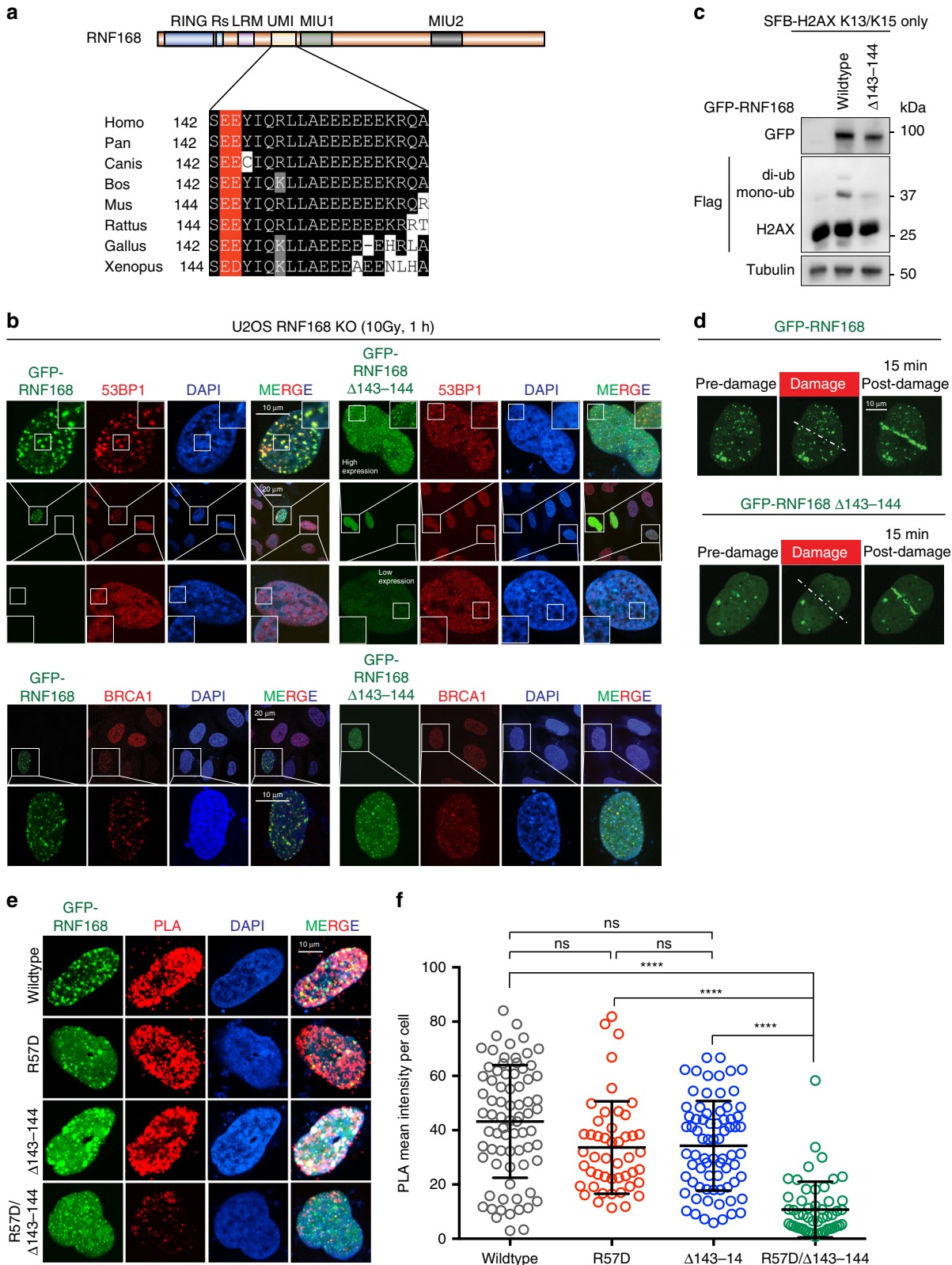

required for generating a specific molecular signal to recruit downstream effector repair proteins through their ubiquitin-binding modules. Therefore, it is necessary for RNF168 to catalyze a highly selective E3 reaction on restricted lysine residues. The nucleosome acidic patch is required for both RING1B/BMI1 and RNF168-mediated ubiquitination; however, it was not clear how RNF168 recognizes its specific target among numerous lysine residues on the nucleosomal surface, particularly on the lysine-rich tails of H2A variants.

The identification of the proximity restriction between the target lysine and the alpha1-extension helix further defines the requirements for RNF168 substrate specificity which will help to identify additional potential substrates, if there are any, for BRCA1, RAP80 and RAD18 recruitment to damaged chromatin.

**Fig. 7 UMI acidic residues are required for RNF168-mediated site-specific ubiquitination and the DDR. a** Sequence alignment analysis of RNF168 UMI domains across species. The evolutionary conserved E143/E144 are highlighted in red. **b** Deletion of residues 143-144 attenuates 53BP1 IRIF formation. U2OS RNF168 KO cells were transiently transfected with GFP-RNF168 or RNF168 Δ143-144. Cells were irradiated with 10 Gy, allowed to recover for 1 h, followed by immunofluorescence analysis with indicated antibodies. Repeated three times with similar results. **c** Deletion of 143-144 abolishes RNF168-mediated ubiquitination. Co-transfection of GFP-RNF168 or mutant with Flag-H2AX K13/K15 only vector in HEK293T cells followed by western blot analysis. Repeated three times with similar results. **d** RNF168 Δ143-144 is recruited to laser damage. U2OS RNF168 KO cells expressing GFP-RNF168 and RNF168 Δ143-144 were damaged using a 405 nm laser source and analyzed 15 min later by confocal microscopy. Dotted line indicates the laser path. Repeated three times independently with similar results. **e-f** U2OS cells were transfected with GFP-RNF168 wildtype and mutants. Proximity ligation assay (PLA) was performed using specific antibodies against GFP and γH2AX. Mean intensity of PLA signals in GFP-positive cells were measured and analyzed using ImageJ software. Data presented as mean ± SD. One-way ANOVA, Dunn's multiple comparison test was used in statistical analysis, wildtype vs. R57D $P = 0.1455$, wildtype vs. Δ143-144 $P = 0.1589$, R57D vs. Δ143-144, $P > 0.9999$, ****$P < 0.0001$ with 95% confidence interval. $n = 75$ (wildtype), 48 (R57D), 74 (Δ143-144), 53 (R57D/Δ143-144) over 3 independent experiments. Source data are provided as Source Data file.

It will also provide us with an additional target to chemically inhibit the DDR pathway through blocking RNF168-mediated ubiquitination without perturbing the acidic patch which may be essential for other cellular functions including RING1B-mediated H2A ubiquitination and H4 histone tail binding. Our result also illuminates the specificity of the E3 ligase catalytic reaction, which provides insight into how other E3 ligases identify their substrates. Intriguingly, a RNF168 homolog is not found in yeast, as it arose later during evolution. Human RNF168 does not ubiquitinate yHTA and Crb2, the 53BP1 yeast orthologue, which does not contain the UDR binding motif that recognizes the H2A K15 ubiquitination. These observations suggest that the whole axis of RNF168-mediated 53BP1 DSB recruitment was not present in early evolution.

Due to the lack of structural information on the full-length RNF168, previous studies using biochemical or computational structural analyses on the RNF168 regulatory mechanism are largely restricted on the RING domain and fragments of structural domains[13,41–43,51]. Interestingly, a previous study showed that RNF168 residues 1-189 exhibits significantly higher activity even though residues 1-113 are sufficient to ubiquitinate H2A(X) in vitro[42]. Moreover, RNF168[−/−] MEF cells with the reconstitution of RNF168 Δ61-168 (LR and UIM motifs) failed to form 53BP1 IRIF[52]. These data are in line with our finding that full RNF168 activity requires more regulatory/promoting elements beyond the RING finger domain. The alpha1-extension helix has two serine and two arginine residues. We predict that this structural region may act as a "reverse" arginine anchor by interacting with the RNF168 acidic residue-rich region between residues 96-177. A detailed structural analysis is needed to confirm this speculation. The majority of our deletion mutants within the linear alpha-helix of RNF168 LRM-UMI-MIU1 motifs did not affect 53BP1 IRIF formation, suggesting that they are unlikely to alter the protein structure and stability that impact RNF168 activity. Within the acidic region between residues 96-177, the UMI hydrophilic surface residues E143/E144 are required for RNF168-mediated ubiquitination of H2A variants and 53BP1 IRIF formation. Interestingly, RNF168 Δ143-144 is still able to localize at DSBs formed by laser-induced micro-irradiation even though its IRIF formation is diminished. This IRIF defect is possibly caused by the attenuation of H2A and H2AX K13/15 ubiquitination due to the loss of RNF168 specificity and its MIU2 motif-dependent positive feedback loop at DNA damage sites. Supporting the importance of these two acidic amino acids, sequence alignment analysis showed the highest conservation on E143/E144 across RNF168 orthologues compared to other acidic regions within the LR, UMI, and MIU1 motifs (Fig. 7a and Supplementary Fig. 7a).

We have ruled out the possibilities that E143/E144 deletion alters RNF168 DSBs recruitment (Fig. 7d) or impedes the intrinsic E3 ligase activity as the RING domain alone displays high E3 activity[8]. Based on the structural analysis, the E143/E144 resides on the hydrophilic side of the helix, implicating that it may not be directly involved in ubiquitin binding[16,51]. We surmised that these two negatively charged acidic residues are responsible for structural recognition and directing RNF168 target specificity, possibly at the positively charged region of the nucleosome at the N-terminus of H2A and C-terminus of H2B. However, we do not exclude the possibility that these two residues may be required for stabilizing the E3-E2 conformation and ubiquitin conjugation.

Our working model highlights the complexity of DDR pathway regulation through the actions of RNF168 at damaged chromatin. RNF8 substrates, histone 1 or L3MBTL2[15,53], recruit RNF168 to damaged chromatin via binding to the LR-UMI-MIU1 (ubiquitin-dependent DSB recruitment module 1-UDM1) motifs. The hydrophilic residue of the UMI motif may be involved in orienting RNF168 on the nucleosome and triggering the discharge of ubiquitin from the E2 conjugating enzyme precisely onto the lysine residue of the histone H2A variants. The MIU2-LRM2 (UDM2) motifs bind to the RNF168-K13/K15 ubiquitination mark and promote the amplification of this mark. We speculate that the K13/K15 ubiquitination may shift the binding preference from the RNF168 UMI motif to the MIU2 motif in order to spatially control RNF168 on damaged chromatin in close proximity to an unmodified nucleosome and initiate an ubiquitination chain reaction surrounding the break site. Similar to our result, a previous report showed that overexpression of RNF168 N-terminus 1-220 is sufficient to trigger H2A ubiquitination and restore 53BP1 IRIF formation in RNF168 deficient cells suggesting that RNF168 is a limiting factor at DSB sites while ectopic expression of the catalytic unit can override the inability to form IRIF[54]. This observation also suggests that RNF168 MIU2 functions solely in focal accumulation via the positive feedback loop. In addition, although in vitro data showed that RNF168 1-113 is sufficient to catalyze H2A(X) ubiquitination in the context of the nucleosome, the LR-UMI-MIU1 motifs are indispensable for the biological function of RNF168 in the DDR pathway. In-depth structural analyses of RNF168 1-190 and nucleosome are needed to provide detailed insight into how RNF168 E143/E144 spatially interacts with the nucleosome and how the H2A alpha1-extension helix is involved in RNF168-specificity.

Interestingly, the alpha1-extension helix is not only important for RNF168-mediated ubiquitination, but also for RNF169 and 53BP1 UDR recognition[10,55]. Although RNF168 is able to ubiquitinate both K13 and K15 residues on H2A and H2AX, only the K15 ubiquitination is recognized by the 53BP1 UDR motif[10]. This interaction specificity seems to involve the sequence around the RNF168-targeted lysine K15 residue. The position of the R17 residue on the alpha1-extension helix that flanks the ubiquitinated lysine is required for ub-H2A and 53BP1 UDR interaction. This observation suggests that both ubiquitinated H2AZ and macroH2A1/2 are not able to interact with 53BP1.

Emerging evidence has revealed that H2AZ and macroH2A are involved in DNA repair[23,24,29]. Our data showed that these RNF168-targeted ubiquitinations of H2A variants are DNA damage-dependent, which may be important to the RNF168-mediated recruitment of DNA repair proteins to DSBs. A detailed characterization of the functional impact of RNF168-mediated H2AZ and macroH2A1/2 ubiquitination in the DDR pathway using a genetic approach will provide us with a definitive answer to how RNF168-histone variants dictate DSB repair. Our working model proposes that the nucleosome acidic patch and the H2A variant alpha1-extension helix may form a molecular clamp to orient RNF168 directionally to the target lysine residue. Taken together, our findings reveal new RNF168 substrates and unravel a regulatory mechanism of the RNF168-orchestrated DDR pathway.

## Methods

**Cell culture**. HEK293T and U2OS cells were purchased from American Type Culture Collection and cultured in Dulbecco's modified Eagle medium (DMEM) with 10% fetal bovine serum supplemented with 100 U/mL penicillin and 100 µg/mL streptomycin at 37 °C and 5% $CO_2$. Transfections were carried out using both Polyethylenimine (PEI) (Polysciences, Inc.) and FuGene HD (Promega) according to the manufacturer's instruction. U2OS RNF168 KO cells[56] were generated using CRISPR/Cas9 method. RNF168 gRNA1–GCATAAACTCGCCTTTTCGA and gRNA2–GGAAGTGGGT GAGTAACCA were cloned into pSpCas9(BB)-2A-Puro (a gift from Fen Zhang–Addgene ID: 48139). RNF168 knockdown in HEK293T cells was performed using Lipofectamine ® RNAiMax (Invitrogen) according the manufacturer's instruction with previously verified siRNAs (#1: 5'-GGCGAAGAGCGAUGGAAGA-3'; #2: 5'-GACA-CUUUCUCCCACAGAUA-3')[5,9] (Horizon Discovery).

**Plasmids and molecular cloning**. Human H2AX and RNF168 open reading frames were used previously[41]. Human H2AZ, macroH2A1, macroH2A2, and H2A.Bbd Gateway-cloning compatible pENTRs were obtained from Harvard PlasmidID Database. Yeast HTA (yHTA) cDNA was amplified by PCR from the *Saccharomyces cerevisiae* strain BY4741 genomic DNA. The cDNAs were subcloned into expression vectors harboring N-terminal SFB (S-protein, Streptavidin binding peptide and Flag-epitope)–tag with either CMV or EF1α promoter, Myc-epitope tag or GFP-tag using Gateway cloning technology (Invitrogen). Gateway reactions were performed using 1 µL insert plasmid, 1 µL empty backbone, 1 µL BP or LR clonase enzyme mix, and 2 µL TE, then allowed to sit for 1 h at room temperature before transforming into TOP10 or DH5α competent cells. Mutations were created using the QuikChange site-directed mutagenesis kit (Agilent Genomics) or the Q5 site-directed mutagenesis kit (New England BioLabs) according to the manufacturer's instructions. Mutagenesis primers were obtained through Integrated DNA Technologies or Sigma Aldrich. cDNA and mutagenesis were verified by Sanger sequencing.

**Antibodies**. Primary antibodies used in this study were Flag M2 (Sigma, F1804; 1:3000 for blotting), Myc (Santa Cruz, sc-40; 1:1000 for blotting), GFP (Invitrogen, A11122; 1:1000 for blotting), 53BP1 (Novus Biologicals, NB100-304; 1:500 for immunofluorescence), BRCA1 (Santa Cruz Biotechnology, SC-6954; 1:500 for immunofluorescence), γH2AX (Millipore, 05-636; 1:1000 for blotting, 1:500 for immunofluorescence), tubulin (Abcam, AB 6046; 1:5000 for blotting), FK2 (Millipore, 04-263; 1:500 for immunofluorescence), actin (Santa Cruz, sc-1616; 1:5000 for blotting), H2AZ (Cell Signaling, 2718; 1:1000 for blotting), MacroH2A1 (Abcam, ab37264; 1:1000 for blotting) and MacroH2A2 (Sigma Aldrich, HPA035865; 1:1000 for blotting). For western blotting, secondary antibodies HRP-linked anti-rabbit IgG and HRP-linked anti-mouse IgG were purchased from Cell Signaling (0704 and 0706; 1:1000). For immunofluorescence, Alexa Fluor 488 goat anti-rabbit and Alexa Fluor 594 goat anti-mouse antibodies were used (Invitrogen; 1:500), DAPI (Thermo Fisher Scientific; 1:2000).

**Western blotting**. Cells were transfected with SFB-tagged, Myc-tagged, and GFP-tagged plasmids as indicated and harvested in 1× Laemmli sample buffer, resolved by SDS-PAGE, transferred to PVDF membranes, immunoblotted with antibodies as indicated and imaged using BioRad ChemiDoc MP.

**In vitro ubiquitination assay**. H2AZ-containing nucleosome core particle (NCP) was obtained from Active Motif ® (81072). MacroH2A octamer and 601 DNA (207 bp) were obtained from Colorado State University Histone Source. MacroH2A NCP was reconstituted by the step dilution method[57] by mixing the 601 DNA with octamer in a 1:1.2 ratio. NCP formation was confirmed with 6% native TBE gels.

In vitro ubiquitination assays were performed using 2.5 µg of recombinant mononucleosomes. NCPs were incubated in a 50 µL reaction buffer containing 50 mM Tris-HCl, pH 7.5. 100 mM NaCl, 10 mM $MgCl_2$, 1 µM ZnOAc, 1 mM DTT, 30 nM ubiquitin-activating enzyme E1 (Boston Biochem), 1.5 µM ubiquitin-

conjugating enzyme UbcH5a (Boston Biochem), 22 µM ubiquitin (Boston Biochem), 3.33 mM ATP and 4 µM RNF168 (1-113) at 30 °C for overnight. The reaction was stopped by adding 1× Laemmli sample buffer. Ubiquitination was analyzed by western blot.

**Immunofluorescence and confocal microscopy**. Cells were seeded on poly-L-lysine coated coverslips (BD biosciences) 24–48 h prior to the experiment. Coverslips were washed in PBS and fixed in 1 mL of 3% paraformaldehyde for 15 min at room temperature followed by a PBS wash then permeabilized with 1 mL of 0.5% Triton-X-100 solution. For FK2 staining, cells were pre-extracted with 0.5% Triton-X 100 solution for 10 s prior to fixation. Samples were then washed again in 1× PBS and incubated with primary antibodies (1:500 in 3% BSA) for 1 h, followed by a PBS wash. Coverslips were then incubated with secondary antibodies (1:500 in 3% BSA) and DAPI (200 µg/mL) for 30 min at room temperature protected from light. Samples were then mounted onto glass slides with an anti-fade solution (0.02% p-phenylenediamine [Sigma, P6001] in 90% glycerol in PBS). Samples were visualized and captured using a Ti-2 inverted C2 + confocal microscope.

**Laser-induced micro-irradiation**. U2OS cells were transfected with GFP-expression vectors as indicated. Twenty-four hours prior to the experiment, cells were seeded on 35 mm glass-bottom dishes and sensitized with 10 µM BrdU. Laser-induced micro-irradiation was performed using a Nikon Ti-2 inverted fluorescent microscope and C2 + confocal system. Cells were damaged with a fixed-wavelength (405 nm) laser at 60% power. Live cell images were recorded in 1-min intervals after damage.

**Sequence alignment, illustration, and molecular graphics**. Sequence alignment was performed using Clustal Omega and illustrated using BoxShade (v. 3.21). All molecular graphics were created using PyMOL. Protein data bank (PDB) files were obtained from a publicly available source–https://www.rcsb.org. H2A-containing nucleosome (PDB-id: 6FQ5); H2AZ-containing nucleosome (PDB-id:5Z30); MacroH2A-containing nucleosome (PDB-id:2F8N). Same PDB files were used for acidic patch illustration (Fig. 3b), H2A variants N-terminal tails superimposition (Fig. 4b-e), H2A variants nucleosomes superimposition (Fig. 5a) and nucleosome electrostatic potential analyses (Fig. 5d). Electrostatic potential was calculated using Adaptive Poisson-Boltzmann Solver and the AMBER force field. The 13 independent Human H2A-containing nucleosomes PDB files (6O1D, 5GXQ, 6IQ4, 4Z5T, 5Y0C, 6JR0, 5AVC, 5ZBX, 6KVD, 5B0Z, 6IPU, 3AV2, and 2CV5) and 5 H2A-containing nucleosome PDB files (1AOI, 5B1M, 2CV5, 1EQZ and 6PWE) from different species were used for superimposition and H2A N-terminal tails analysis (Supplementary Fig. 5a-f).

**Proximity ligation assay**. The Duolink® In Situ Red Starter kit (Sigma, DUO92101) for mouse/rabbit antibodies was used for proximity ligation assays. Cells were transfected with GFP-tagged plasmids as indicated and seeded onto coverslips 24 h prior to experiment. Coverslips were then irradiated, fixed and treated following the manufacturer's instructions. Briefly, after incubation with Duolink® PLA mouse and rabbit probes, coverslips were treated with antibodies against GFP and γH2AX at 1:500 dilutions. Proximity ligation signal was visualized and captured using a Ti-2 inverted C2 + confocal microscope. Data analysis was performed by measuring the mean fluorescence intensity of the TRITC channel using ImageJ (v.1.51 s) software. Statistical analysis was performed using GraphPad Prism 6 by One-way ANOVA with Dunn's multiple comparison test. Data are represented as mean ± S.D. as indicated.

**Reporting summary**. Further information on research design is available in the Nature Research Reporting Summary linked to this article.

## Data availability
The source data Figs. (1b-e, 2b-i, 3c-e, 4g-k, 5c, 7c, 7f) and Supplementary Figs. (1a-e, 3a-c, 4b) are provide as Source Data file. All of our Protein Data Bank (PDB) files were obtained from RCSB PDB (Research Collaboratory for Structural Bioinformatics Protein Data Bank)–http://www.rcsb.org. All datasets generated in the current study are available from the corresponding author upon reasonable request.

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

## Acknowledgements

We thank Drs. Fen Xia (UAMS), Tanya Paull (UT Austin), Ilya Finkelstein (UT Austin), and John Pehrson (U Penn) for the insightful discussion. We would also like to thank Alicia Byrd (UAMS), Drs. Uma Muthurajan and Hataichanok Scherman (Colorado State University Histone Source) for the technical supports on nucleosome reconstitution, the UAMS OSPAN Science Communication (SciCom) group for editing the manuscript. This work was supported by grants from the NIH (K22CA204354) and Arkansas Breast Cancer Research Program (AWD00053730) to J.W.L. and in part by start-up funds from the University of Arkansas for Medical Sciences and NIH (P20GM121293). The National Natural Science Foundation of China (Grant 31770805) to Q.G.

## Author contributions

J.W.L. conceptualized the study, J.K., K.W., Q.G., J.W.L. performed experiments, J.K., J.W.L. wrote the manuscript, K.W., Q.G. provided intellectual contribution throughout the project. J.W.L. supervised the study.

## Competing interests

The authors declare no competing interests
