## [Peer Review File · Nature Communications]

Reviewers' Comments:

Reviewer #1:

Remarks to the Author:

In the manuscript titled "Histone H2A variants alpha1-extension helix directs RNF168-mediated ubiquitination", the authors showed some interesting findings that RNF168 ubiquitinates several non-canonical H2A variants (H2AZ and macroH2A1/2) at the divergent N-terminal tail lysine residue. Further, they also identified that the acidic patch and N terminal alpha 1-extension helix of H2A (H2A variants), as well as UDM1 and UIM motifs of RNF168 are important to RNF168 mediated H2A (H2A variants) mono-ubiquitination.

Generally, these studies showed new substrates of RNF168 and identify new motifs important for RNF168-mediated reaction, which may be important to DNA double-strand break repair. The data are mostly solid and of interest to the field. However, there are still several concerns.

1. The authors show that several non-canonical H2A variants can be ubiquitinated at some specific sites by RNF168. So, the question is what is the functional significance of these RNF168 mediated ubiquitination? At least, this needs to be discussed.
2. To confirm whether H2A variants are really ubiquitinated by RNF168, the in vitro Ub assays are also necessary.
3. Is RNF168 mediated H2A variant (H2AZ and macroH2A1/2) ubiquitination also DNA damage-induced?
4. The unequal expression of RNF168, such as Fig 3e, 4e makes the results hard to interpret.

Minor:

1. In Fig 1b, 1c, 2b, 2c, 2e, et al, the western blot of Flag-H2AZ, H2AX and MacroH2A always show multi-bands, the authors should indicate which bands are the non-ubiquitinated bands or unspecific bands? Which bands are the ubiquitinated bands? Otherwise, it is easy to be confused.
2. In Fig 2c, what did the authors detect in the top panel? Myc-RNF168?

Reviewer #2:

Remarks to the Author:

This manuscript describes a series of carefully-executed experimentations to identify molecular determinants that promote RNF168-dependent histone ubiquitylation events. The authors showed that RNF168 mediates ubiquitylation of H2A variants, including H2AZ and macroH2A1/2, and revealed important roles of the acidic patch as well as the alpha1-extension helix in supporting these ubiquitylation reactions.

While these findings may be important to further our understanding of the E3 ligase RNF168 and its broader role as a chromatin modifier, it is at present unclear if H2AZ and macroH2A1/2 are bona fide RNF168 substrates, and whether their respective ubiquitin adducts may also serve as docking platforms for 53BP1. In a revised manuscript the authors should also provide an adequate introduction on the various H2A variants including their chromatin distribution and their participation in the DDR.

Major Comments:

- 1) Most of the histone ubiquitylation experiments were performed in over-expression settings. The authors should provide direct evidence that the histones are ubiquitylated in vivo.
- 2) At present there is not sufficient evidence to support that RNF168 catalyses ubiquitylation of H2AZ and macroH2A1/2. The authors should perform in vitro ubiquitylation experiments to support the claim, ideally with nucleosomal histones.
- 3) Do H2AX RAG/AAA or SRS/AAA or SRSSRAG/AAASAAA support 53BP1 IRIF in H2AX null cells?
- 4) Does over-expression of the H2AZ or the macroH2A1/2 mutants defective in RNF168-mediated ubiquitylation affect 53BP1/FK2 IRIF?

Minor Comment - The text is in need of extensive editing to improve on clarity and flow. Some of the images are also of sub-standard quality (e.g. Figure 6b & c).

Reviewer #3:

Remarks to the Author:

RNF168 is an important ubiquitin E3 ligase that targets the N-terminus of H2A, and is important in double strand break repair. Here Kelliher et al. report on their investigation of RNF168 ubiquitination of H2A. Specifically, they focus on the ubiquitination of H2A variants H2AZ and macroH2A1/2 and the structural basis of specificity. Unfortunately, the study falls short in several areas, and needs substantial modification before it would be ready for publication.

Major points:

- 1) The identification of H2AZ as a new target of RNF168 is incorrect as this has been identified previously (PMC4693525). This previous result should be discussed and these results put in that context.
- 2) The assay for determining that H2AZ and macroH2A1/2 are targets of RNF168 is not fully convincing. Both are overexpressed at high levels in 293T cells, which could lead to many off target effects. The direct targeting of these proteins by RNF168 needs to be validated
- 3) It is stated that "The mechanism by which RNF168 and RING1B target the specific lysine among the nine lysine residues on the H2A tails or other lysine residues on the nucleosome is not fully understood". However, a recent paper by Horn et al. (PMCID: PMC6465349), which is only mentioned in passing, carried out an extensive analysis of the specificity of ubiquitination of H2A by RNF168.
- 4) The authors state that the acidic patch interaction is not enough, but their own data (as well as others) show that mutation of this interface almost completely abolishes ubiquitination of H2A, so this is not clear.
- 5) The authors identify a conserved patch in the "alpha1 extension helix" of H2A and suggest that this may be important for specificity by orienting RNF168. However, this is never tested. Though mutation of these residues is shown to abrogate ubiquitination, it is entirely unclear what role these residues are playing. The same can be said for the two glutamic acids that they identify in RNF168. As is, this is purely speculative.

Minor points:

- 1) English needs editing throughout
- 2) The figure legends are not sufficient and should be expanded to fully notate the figures themselves.
- 3) Blots throughout are very poorly marked. Bands should be clearly marked and labels explained more clearly. E.g. does K4 only mean that only K4 is present and all the rest of the lysines are arginines? Or does this mean that only K4 was mutated to arginine? What band is the Flag notation marking? What are the other bands?
- 4) It is suggested that the structure of the N-terminal tails is different. However as depicted it is not clear how they differ with respect to the nucleosome core. Please add the DNA back in for clarity and expand on what this structural difference is.
- 5) The PDBid for the H2AZ nucleosome is incorrect.

Reviewer #1 (Remarks to the Author):

In the manuscript titled “Histone H2A variants alpha1-extension helix directs RNF168-mediated ubiquitination”, the authors showed some interesting findings that RNF168 ubiquitinates several non-canonical H2A variants (H2AZ and macroH2A1/2) at the divergent N-terminal tail lysine residue. Further, they also identified that the acidic patch and N terminal alpha 1-extension helix of H2A (H2A variants), as well as UDM1 and UIM motifs of RNF168 are important to RNF168 mediated H2A (H2A variants) mono-ubiquitination.

Generally, these studies showed new substrates of RNF168 and identify new motifs important for RNF168-mediated reaction, which may be important to DNA double-strand break repair. The data are mostly solid and of interest to the field. However, there are still several concerns.

We would like to thank the reviewer for the detailed evaluation and for finding our work potentially important for the scientific community. We also appreciate the critical comments, especially the suggestions for improving the study. We gave our best to improve the manuscript and added a substantial amount of data, including knockdown studies and a potential connection of DNA damage in H2A variants ubiquitination.

1. The authors show that several non-canonical H2A variants can be ubiquitinated at some specific sites by RNF168. So, the question is what is the functional significance of these RNF168 mediated ubiquitination? At least, this needs to be discussed.

We have performed additional experiments to show that the RNF168-targeted ubiquitination in H2AZ and macroH2A1/2 are DNA damage-induced (Supplementary Fig. 3). We have included substantial discussion in the manuscript on the potential biological function in the context of DDR.

2. To confirm whether H2A variants are really ubiquitinated by RNF168, the in vitro Ub assays are also necessary.

Thank you for the suggestion. We have performed in vitro ubiquitination assay using commercially available H2AZ-containing nucleosome (Supplementary Fig. 1). We have also reconstituted macroH2A-containing nucleosome and performed an in vitro ubiquitination assay (Fig. 1c-e). We confirmed that both H2AZ and macroH2A are bona fide RNF168 substrates.

3. Is RNF168 mediated H2A variant (H2AZ and macroH2A1/2) ubiquitination also DNA damage-induced?

This is an interesting point. Majority of the H2A ubiquitination is not RNF168-dependent, thus the change of RNF168-specific H2A ubiquitinations is difficult to measure upon DNA damage. In order to examine whether the RNF168-mediated H2A variants ubiquitination is DNA damage-induced, we have established stable cell lines expressing mutant H2A variants which has both histone tails lysines mutated to arginine (K to R)

except the RNF168-targeted lysine (H2AZ K15, macroH2A1 K11, macroH2A2 K11) in HEK293T background. These genetic reagents allow us to measure the RNF168-target in response to DNA damage and avoid other lysine ubiquitinations masking the RNF168-mediated ubiquitination. We exposed the cells to radiation and probed for ubiquitinated bands using flag antibody. In order to avoid confusion, we labeled them H2AZ 9K to R-R15K, macroH2A1 8K to R-R11K, macroH2A2 8K to R-R11K. Our data showed that the RNF168-targeted lysine ubiquitinations in H2A variants are DNA damage-induced (Supplementary Fig. 3a-c).

4. The unequal expression of RNF168, such as Fig 3e, 4e makes the results hard to interpret.

Thank you for pointing this out. We have re-performed the experiment and updated the figures.

Minor:

1. In Fig 1b, 1c, 2b, 2c, 2e, et al, the western blot of Flag-H2AZ, H2AX and MacroH2A always show multi-bands, the authors should indicate which bands are the non-ubiquitinated bands or unspecific bands? Which bands are the ubiquitinated bands? Otherwise, it is easy to be confused.

We apologize for the confusion. We have included labels in each figure to indicate whether they are ubiquitinated bands, non-ubiquitinated bands and non-specific bands.

2. In Fig 2c, what did the authors detect in the top panel? Myc-RNF168?

We have added the label in the figure.

Reviewer #2 (Remarks to the Author):

This manuscript describes a series of carefully-executed experimentations to identify molecular determinants that promote RNF168-dependent histone ubiquitylation events. The authors showed that RNF168 mediates ubiquitylation of H2A variants, including H2AZ and macroH2A1/2, and revealed important roles of the acidic patch as well as the alpha1-extension helix in supporting these ubiquitylation reactions.

While these findings may be important to further our understanding of the E3 ligase RNF168 and its broader role as a chromatin modifier, it is at present unclear if H2AZ and macroH2A1/2 are bona fide RNF168 substrates, and whether their respective ubiquitin adducts may also serve as docking platforms for 53BP1. In a revised manuscript the authors should also provide an adequate introduction on the various H2A variants including their chromatin distribution and their participation in the DDR.

We appreciate that the reviewer found our work interesting and important. We would also like to thank the reviewer for the comments and suggestions. We have included

background information of H2A variants, their chromatin distribution and their roles in the DDR pathway in the introduction, Moreover, we have consulted our institutional editorial service to extensively edit the manuscript to improve the flow and data presentation. We have also included additional experiments as suggested.

Major Comments:

1) Most of the histone ubiquitylation experiments were performed in over-expression settings. The authors should provide direct evidence that the histones are ubiquitylated in vivo.

Thank you for the suggestion. Similar to H2A/H2AX, RNF168-mediated H2AZ and macroH2A1/2 ubiquitinations are very low endogenously. This is why we use the same approach as previous reports^{1,2} (Mattioli, F. et al, 2012; Leung, JW. et al, 2014) to study the RNF168-catalyzed site-specific ubiquitination. We agree that there is a potential concern on the indirect effects of ectopic over-expression. To this end, we performed additional experiments by depleting RNF168 in HEK293T cells stably express H2AZ 9K to R-R15K, macroH2A1 8K to R-R11K or macroH2A2 8K to R-R11K. Since the endogenous RNF168-mediated ubiquitination is barely detectable, we performed pull-down to obtain a more discernible signal. We did observe RNF168 depletion consistently attenuates H2AZ, macroH2A1 and macroH2A2 ubiquitination (Fig. 2g-i). These data are in line with our co-transfection experiment with lysine mutants (Fig. 2c-f).

2) At present there is not sufficient evidence to support that RNF168 catalyses ubiquitylation of H2AZ and macroH2A1/2. The authors should perform in vitro ubiquitylation experiments to support the claim, ideally with nucleosomal histones.

Thank you for the suggestion. We have included in vitro experiments in figure 1 and supplementary figure 1. Also see review 1, major comment 2.

3) Do H2AX RAG/AAA or SRS/AAA or SRSSRAG/AAASAAA support 53BP1 IRIF in H2AX null cells?

We have reconstituted the H2AX-SRSSRAG/AAASAAA mutant in H2AX KO U2OS cells and it restored 53BP1 similar to the previously reported H2AX K13/15R mutant that is defective for RNF168-mediated ubiquitination² (Supplementary Fig. 4c).

4) Does over-expression of the H2AZ or the macroH2A1/2 mutants defective in RNF168-mediated ubiquitylation affect 53BP1/FK2 IRIF?

We have overexpressed H2AZ K15R, macroH2A1 K11R and macroH2A2 K11R in U2OS cells and perform immunofluorescence experiment to detect 53BP1 and FK2 foci after irradiation and we did not see any change in these mutants compared to wildtype cells. We have included the data in supplementary figure 2.

Minor Comment - The text is in need of extensive editing to improve on clarify and flow.

Some of the images are also of sub-standard quality (e.g. Figure 6b & c).

Thank you for the comment. We have consulted our institutional editorial service to edit our manuscript and we hope the clarity and flow has significantly improved. We have also included high-resolution enlarged image in Figure 6b and 6c.

Reviewer #3 (Remarks to the Author):

RNF168 is an important ubiquitin E3 ligase that targets the N-terminus of H2A, and is important in double strand break repair. Here Kelliher et al. report on their investigation of RNF168 ubiquitination of H2A. Specifically, they focus on the ubiquitination of H2A variants H2AZ and macroH2A1/2 and the structural basis of specificity. Unfortunately, the study falls short in several areas, and needs substantial modification before it would be ready for publication.

We would like to thank the reviewer for the critical comments. We did our best to improve our study by addressing the concerns, which include the RNF168-depletion and proximity assay to support our working model on the potential function of the alpha1 extension helix.

Major points:

1) The identification of H2AZ as a new target of RNF168 is incorrect as this has been identified previously (PMC4693525). This previous result should be discussed and these results put in that context.

Sorry for the oversight. We have included this previous finding in our introduction and discussion.

2) The assay for determining that H2AZ and macroH2A1/2 are targets of RNF168 is not fully convincing. Both are overexpressed at high levels in 293T cells, which could lead to many off target effects. The direct targeting of these proteins by RNF168 needs to be validated

Thank you for the suggestion. To confirm whether RNF168 directly targets H2AZ and macroH2A, we have included RNF168-depletion experiment in cells (Fig 2. g-i). We have also performed in vitro ubiquitination assay in the context of nucleosome, which we have included these data in Fig. 1c-e and Supplementary Fig. 1a.

3) It is stated that “The mechanism by which RNF168 and RING1B target the specific lysine among the nine lysine residues on the H2A tails or other lysine residues on the nucleosome is not fully understood”. However, a recent paper by Horn et al. (PMCID: PMC6465349), which is only mentioned in passing, carried out an extensive analysis of the specificity of ubiquitination of H2A by RNF168.

We apologize for the oversight again. We totally agree with the paper “Structural basis of specific H2A K13/K15 ubiquitination by RNF168” which reported extensive molecular detail on the RNF168-RING domain and nucleosome. In fact, this publication gives us a lot of insights into our experimental design and execution. We have added a comprehensive discussion in the introductory section to discuss the impact of their recent findings and how it connects to our study.

4) The authors state that the acidic patch interaction is not enough, but their own data (as well as others) show that mutation of this interface almost completely abolishes ubiquitination of H2A, so this is not clear.

This is a very interesting point. The acidic patch is absolutely required for RNF168-mediated H2A ubiquitination and the RNF168 R57D mutant attenuates the NCP interaction³. We believe that the acidic patch may play an important role as a scaffold docking to E3 ligases including RNF168 and RING1B. We think there is an additional element to direct RNF168 to the specific lysine targets among many lysines on the nucleosome surface. By mutagenesis, we found that the alpha1 extension helix, which is located in close proximity to the RNF168-targeted lysines, is also required for RNF168-mediated ubiquitination. Our working model propose that the alpha1 extension helix may help precisely orientating RNF168 on the nucleosome surface or stabilizing the E3-E2 conformation to achieve target specificity.

5) The authors identify a conserved patch in the “alpha1 extension helix” of H2A and suggest that this may be important for specificity by orienting RNF168. However, this is never tested. Though mutation of these residues is shown to abrogate ubiquitination, it is entirely unclear what role these residues are playing. The same can be said for the two glutamic acids that they identify in RNF168. As is, this is purely speculative.

Thank you for the comment. We totally agree with it. However, crystallography and structural analysis of RNF168 1-190 and nucleosome is extremely challenging. Due to our technical limitation, we are not able to provide additional data to prove our working model whether RNF168 E143-144 are directly interacting with the nucleosome alpha1-extension helix.

In order to address this important question, we use proximity ligation assay (PLA) to measure the potential distance and interaction between RNF168 and nucleosome (Fig. 7e and f). We expressed GFP-RNF168, GFP-RNF168 R57D, GFP-RNF168 Δ 143-144, and GFP-RNF168 R57D/ Δ 143-144 in U2OS and performed the PLA using anti-GFP (Rabbit) and anti- γ H2AX (mouse). Positive signal represents a close proximity of two targeted molecules. We measured the mean intensity of the PLA signal in each GFP-positive cells and we found GFP-R57D/ Δ 143-144 showed a drastic reduction of PLA signal compared to the wildtype, R57D and Δ 143-144 mutants. These data suggested that both R57D and E143-144 are required to recruit RNF168 to the nucleosome. We have rewritten and rephrased our results and discussion to conclude our scientific discovery accurately. We do believe further in-depth studies are necessary to fully uncover the molecular action between RNF168 and nucleosome.

Minor points:

1) English needs editing throughout

We have consulted our institutional editorial service to edit our manuscript. We hope the flow of the manuscript is improved.

2) The figure legends are not sufficient and should be expanded to fully notate the figures themselves.

We apologize for the lack of detail in the figure legends. We have now added substantial notation to improve the clarity.

3) Blots throughout are very poorly marked. Bands should be clearly marked and labels explained more clearly. E.g. does K4 only mean that only K4 is present and all the rest of the lysines are arginines? Or does this mean that only K4 was mutated to arginine? What band is the Flag notation marking? What are the other bands?

We sincerely apologize for the confusion. We made changes in the labels in the figure, we also added remarks in the text and figure legends to clarify. For lysine to arginine mutants, we used K-R. On the contrary, for the mutant with only ONE lysine only at the histone tail, we adopt the notation we used in our previous publication². We labeled it as e.g. H2AZ 9K to R-R15K to indicate mutations of all 9 lysines (K) on the both histone tails to arginine (R) with the arginine reverse back to lysine at position 15. We have also added labels on each band in our blots, including the non-ubiquitinated band, mono-ubiquitinated band and di-ubiquitinated band (see Review 1, minor comment 1).

4) It is suggested that the structure of the N-terminal tails is different. However as depicted it is not clear how they differ with respect to the nucleosome core. Please add the DNA back in for clarity and expand on what this structural difference is.

Thank you for the suggestion, we have included the DNA in the H2A variants comparison (see Fig. 4b-e), H2A histone tails are generally flexible and dynamic and do not form a structure. In order to clarify, we have made changes in our result and discussion. In Fig. 4b-e (Fig. 2h-l in the first submission), we wanted to illustrate the RNF168-targeted lysines are at the unstructured H2A variants N-terminal tails, where they share low conservation. We also would like to point out that although RNF168 catalyzes site-specific ubiquitination on H2A variants, the target lysines locate at a different position at their histone tails. This observation prompted our hypothesis that a specific sequence is required for RNF168-mediated ubiquitination in the proximity of the target lysine, it also helped us to predict that their non-consensus tails are unlikely to play a role in the RNF168 target specificity.

5) The PDBid for the H2AZ nucleosome is incorrect.

Thank you for pointing it out, we have made the correction.

References

1. Mattioli, F. et al. RNF168 ubiquitinates K13-15 on H2A/H2AX to drive DNA damage signaling. *Cell* **150**, 1182-95 (2012).
2. Leung, J.W. et al. Nucleosome acidic patch promotes RNF168- and RING1B/BMI1-dependent H2AX and H2A ubiquitination and DNA damage signaling. *PLoS Genet* **10**, e1004178 (2014).
3. Mattioli, F., Uckelmann, M., Sahtoe, D.D., van Dijk, W.J. & Sixma, T.K. The nucleosome acidic patch plays a critical role in RNF168-dependent ubiquitination of histone H2A. *Nat Commun* **5**, 3291 (2014).

Reviewers' Comments:

Reviewer #1:

Remarks to the Author:

The authors have fully addressed my concerns. The identification of new substrates of RNF168 and new motifs important for RNF168 function should be of interest to the field. I support the publication of this manuscript.

Reviewer #2:

Remarks to the Author:

The authors have satisfactorily addressed my queries and IMHO the manuscript is acceptable for publication in Nat Commun.

Reviewer #3:

Remarks to the Author:

This is a revision of a previous report in which Kelliher et al. investigate RNF168 ubiquitylation of H2A variants, H2AZ and macroH2A1/2, in DDR. Several of my concerns have been addressed and a substantial amount of data has been added. However, I still have concerns regarding the major conclusion of this manuscript, that H2AZ and macroH2A1/2 are in vivo targets of RNF168. Without that I'm not convinced that this is suitable for publication in Nature Communications.

The in vitro ubiquitylation assays are an excellent addition to the manuscript and clearly show that H2AZ and macroH2A1 can be ubiquitinated by RNF168 in an isolated setting. However, the data to demonstrate that these are in vivo targets of RNF168 are still not fully convincing. The authors nicely showed that overexpressed, tagged H2AZ and macroH2A1/2 are no longer ubiquitinated in an RNF168 knockdown. However, these tagged proteins are likely still present in much greater abundance than endogenous. This is a major concern as RNF168 has other confirmed H2A targets in the cell and thus the levels protein are important. I understand the need to use a tag due to antibody restrictions. However, either these need to be expressed only at endogenous levels, or another approach, perhaps mass spec, could be used to show that endogenous protein is a target in the RNF168 KD.

Minor points)

- 1) A sequence alignment of the H2A variant tails as compared to canonical would be very helpful
- 2) I'm skeptical of the new analysis of the variation in the orientation of the lysine sidechain (Figure 4). My guess is that this can vary quite a bit from structure to structure even with the same H2A. If the authors compare several different structures of the H2A-nucleosome and see that the position of that side-chain is conserved that would make this comparison more convincing.
- 3) The new section titled "RNF168 functionally requires the UDM1 acidic-rich region" is very difficult to read. There appear to be several references missing. In addition, the labels on the nucleosome in Figure 5 don't all appear to be correct (e.g. K118??). The electrostatic analysis of the nucleosome is presented as a rationale for studying the MIU2 motif, but it is never clear how that relates to the data collected.

Reviewer #3 (Remarks to the Author):

This is a revision of a previous report in which Kelliher et al. investigate RNF168 ubiquitylation of H2A variants, H2AZ and macroH2A1/2, in DDR. Several of my concerns have been addressed and a substantial amount of data has been added. However, I still have concerns regarding the major conclusion of this manuscript, that H2AZ and macroH2A1/2 are in vivo targets of RNF168. Without that I'm not convinced that this is suitable for publication in Nature Communications.

The in vitro ubiquitylation assays are an excellent addition to the manuscript and clearly show that H2AZ and macroH2A1 can be ubiquitinated by RNF168 in an isolated setting. However, the data to demonstrate that these are in vivo targets of RNF168 are still not fully convincing. The authors nicely showed that overexpressed, tagged H2AZ and macroH2A1/2 are no longer ubiquitinated in an RNF168 knockdown. However, these tagged proteins are likely still present in much greater abundance than endogenous. This is a major concern as RNF168 has other confirmed H2A targets in the cell and thus the levels protein are important. I understand the need to use a tag due to antibody restrictions. However, either these need to be expressed only at endogenous levels, or another approach, perhaps mass spec, could be used to show that endogenous protein is a target in the RNF168 KD.

Thank you very much for the comments and constructive advice to improve the quality of our manuscript.

We agree with the concern about the protein level expression. We have checked our tagged-H2A variants expression level using antibodies targeting H2AZ, macroH2A1 and macroH2A2. Our data consistently showed that the CMV-SFB-tagged histone variant expressions are comparable to endogenous levels except macroH2A2. To address this, we cloned the macroH2A2 8KR-R11K into EF1 α -SFB-pDEST expression vector which showed similar expression to the endogenous macroH2A2 level (Fig. 2i)

We have run out of our original samples. Therefore, we have re-performed a new set of experiments (Fig 2g-i) and included exogenous and endogenous protein comparison using specific antibodies. We have also included untransfected controls in the data source file to confirm the antibody specificity to the exogenous protein expression.

Minor points

1) A sequence alignment of the H2A variant tails as compared to canonical would be very helpful

We have added a full sequence alignment of all H2A variants as supplementary figure 1.

2) I'm skeptical of the new analysis of the variation in the orientation of the lysine sidechain (Figure 4). My guess is that this can vary quite a bit from structure to structure

even with the same H2A. If the authors compare several different structures of the H2A-nucleosome and see that the position of that side-chain is conserved that would make this comparison more convincing.

Thank you for the suggestion. We have included two comprehensive analyses i) 13 human H2A-containing nucleosomes; and ii) H2A-containing nucleosome from 5 different species, to compare their lysine sidechains. All of the K13/K15 sidechains show consistent positional and proximal superimposition with a slight directional variation. These data further supported the observation on the N-terminal lysines positional difference between H2A, H2AZ and macroH2A.

3) The new section titled “RNF168 functionally requires the UDM1 acidic-rich region” is very difficult to read. There appear to be several references missing. In addition, the labels on the nucleosome in Figure 5 don't all appear to be correct (e.g. K118??). The electrostatic analysis of the nucleosome is presented as a rationale for studying the MIU2 motif, but it is never clear how that relates to the data collected.

We apologize for our oversight. We have revised that specific section and we also added appropriate references.

We have also double-checked and confirmed that the labels on the nucleosome in Figure 5 are correct. The nucleosome PDBs file in our analyses do not contain full histone tails. Therefore, we labeled the amino acid at the most C-terminus to illustrate the H2A variants C-terminal tail positions. We have also compared our NCP (6FQ5) with a previously published NCP structure (1KX5) (Mattioli, F, et al. Nature Communications, 2014). The K118 residues on both PDB files locate at the same position.

Additionally, we have added clarification on how we rationalized our hypothesis on studying UMI motifs, and the reason why we performed systematic mutagenesis screen on the acidic-rich UDM1 region of RNF168, which is based on our observation of the consistent positively charged on the H2A N-terminus proximal region from the electrostatic potential analysis.

Reviewers' Comments:

Reviewer #3:

Remarks to the Author:

In this resubmission the authors have addressed all of the points I raised in the previous review.

Response to the referees

REVIEWERS' COMMENTS:

Reviewer #3 (Remarks to the Author):

In this resubmission the authors have addressed all of the points I raised in the previous review.

We would like to thank you again for the insightful comments and critiques.